# Correlation between Type I Interferon Associated Factors and COVID-19 Severity

**DOI:** 10.3390/ijms231810968

**Published:** 2022-09-19

**Authors:** Dóra Bencze, Tünde Fekete, Kitti Pázmándi

**Affiliations:** 1Department of Immunology, Faculty of Medicine, University of Debrecen, 1 Egyetem Square, H-4032 Debrecen, Hungary; 2Doctoral School of Molecular Cell and Immune Biology, University of Debrecen, 1 Egyetem Square, H-4032 Debrecen, Hungary

**Keywords:** plasmacytoid dendritic cell 1, type I interferon, COVID-19, SARS-CoV-2, antiviral response, risk factor, IFN signature

## Abstract

Antiviral type I interferons (IFN) produced in the early phase of viral infections effectively inhibit viral replication, prevent virus-mediated tissue damages and promote innate and adaptive immune responses that are all essential to the successful elimination of viruses. As professional type I IFN producing cells, plasmacytoid dendritic cells (pDC) have the ability to rapidly produce waste amounts of type I IFNs. Therefore, their low frequency, dysfunction or decreased capacity to produce type I IFNs might increase the risk of severe viral infections. In accordance with that, declined pDC numbers and delayed or inadequate type I IFN responses could be observed in patients with severe coronavirus disease (COVID-19) caused by the severe acute respiratory syndrome coronavirus 2 (SARS-CoV-2), as compared to individuals with mild or no symptoms. Thus, besides chronic diseases, all those conditions, which negatively affect the antiviral IFN responses lengthen the list of risk factors for severe COVID-19. In the current review, we would like to briefly discuss the role and dysregulation of pDC/type I IFN axis in COVID-19, and introduce those type I IFN-dependent factors, which account for an increased risk of COVID-19 severity and thus are responsible for the different magnitude of individual immune responses to SARS-CoV-2.

## 1. Introduction

Coronavirus disease (COVID-19) caused by a single-stranded RNA virus, the severe acute respiratory syndrome coronavirus 2 (SARS-CoV-2) came from the Chinese city of Wuhan in 2019, and caused pandemic around the world [1]. Since the beginning of the pandemic, more infectious novel virus variants have emerged and led to newer waves of the pandemic [2].

The symptoms of COVID-19 can range from asymptomatic to mild and to severe symptoms in humans indicating that the individual’s immunological state greatly influences the course and outcome of the disease [3].

In general, the course of COVID-19 can be divided into 3 distinct stages based on the clinical manifestation and recommended therapy [4]. The first stage is the early phase of infection, which begins immediately after infection and includes an incubation period. This stage is usually asymptomatic or is associated with mild and often non-specific symptoms such as malaise, fever and dry cough. In patients in whom COVID-19 is restricted to this stage, the prognosis is excellent. The second stage is already associated with pulmonary involvement, which can occur without or with hypoxia. Patients develop viral pneumonia with cough, fever, and possibly hypoxia. At this stage, most patients with COVID-19 require hospitalization for close monitoring or treatment. The severe third stage of the disease manifests as an extrapulmonary, systemic inflammation resulting in a cytokine storm. At this stage, lymphopenia may develop and a decrease in helper, regulatory, and memory T cell counts [5], an increase in neutrophil counts and a significant increase in inflammatory cytokines and biomarkers can be observed [5,6]. In patients with this advanced stage of the disease, a cytokine profile resembling secondary haemophagocytic lymphohistiocytosis may be observed, which is characterized by elevated levels of IL-2, IL-7, granulocyte colony stimulating factor (G-CSF), IFN-γ inducible protein 10 (IP-10), monocyte chemoattractant protein 1 (MCP-1), macrophage inflammatory protein-1α (MIP-1α) and tumor necrosis factor-α (TNF-α) [7]. Furthermore, septic shock, vasoplegia, acute respiratory distress syndrome, cardiopulmonary collapse, as well as systemic organ involvement or even myocarditis may occur at this stage [4]. Overall, the prognosis in this phase of the disease is rather poor, with only a few patients recovering from the critical stage of the disease.

Besides the well-known respiratory pathology, various extrapulmonary manifestations of COVID-19 have already been reported highlighting the involvement of cardiovascular, genitourinary, gastrointestinal and central nervous system as well as the skin [8]. The multi-organ involvement can be manifested by various symptoms including thrombotic complications, myocardial dysfunction, arrhythmia, acute coronary syndrome, acute kidney injury, gastrointestinal symptoms, hepatocellular injury, hyperglycemia and ketosis, neurologic illnesses, ocular symptoms, dermatologic complications, preeclampsia and fertility problems [8,9,10,11,12,13].

In addition, those patients who recovered from COVID-19 may suffer from post-COVID-19 syndrome, which negatively affects their quality of life for months after recovery. The post-COVID-19 syndrome is characterized by a wide variety of clinical symptoms including pulmonary embolism, deep vein thrombosis, acute myocardial infarction, depression, anxiety, myalgia, dyspnea, fatigue, defects in memory and concentration and a variety of neuropsychiatric syndromes [14,15,16,17].

In general the incidence of post-COVID-19 syndrome is about 10–35%; however, this rate can reach up to 85% for those patients who required hospitalization during acute SARS-CoV-2 infection [15].The severity and mortality of COVID-19 is higher in patients with chronic conditions such as diabetes, hypertension, and cardiovascular diseases [18]. A growing body of evidence indicates that individuals with disturbed antiviral interferon (IFN) response are more likely to develop severe COVID-19 symptoms. So far, it seems that mortality rates are higher in seniors, men, pregnant women, and obese patients [19,20,21,22] that might be explained by the impaired or dysregulated type I IFN response, which is a vital component of antiviral immunity. In addition, the pandemic does not spare the young with no underlying medical conditions either that might be related to genetic defects in the IFN signaling pathways or autoantibodies generated against type I IFNs, which neutralize the direct inhibitory effect of type I IFNs on viral replication [23]. Therefore, the individual’s type I IFN signature greatly contributes to the variability of COVID-19 outcome, and conditions associated with decreased type I IFN production indicate poorer prognosis. The main sources of type I IFNs upon viral infections are plasmacytoid dendritic cells (pDCs), which are specialized for the recognition of viral nucleic acids and subsequent release of huge amount of type I IFNs [24]. pDCs and pDC-derived IFNs are central players in the antiviral immune responses against SARS-CoV-2, thus a number of excellent papers have already reviewed the importance of antiviral IFNs [25] and pDCs [26] in COVID-19.

Therefore, in this review, we aimed to briefly summarize the role of IFNs and pDCs in COVID-19 based on the newest available data on this field. Nevertheless, we want to give a deep insight into those risk factors for COVID-19 severity, which are associated with impaired type I IFN responses and reduced pDC number to highlight that low type I IFN signature of individuals due to different inborn or acquired conditions predicts a more severe disease outcome.

## 2. The Role of Antiviral IFNs in COVID-19

SARS-CoV-2 is a positive-stranded RNA virus, which mainly infects via the respiratory tract. The first targets of the virus are the most permissive ciliated airway epithelial cells in the nasal mucosa. The virus uses the angiotensin-converting enzyme 2 (ACE-2) receptor and the transmembrane serine protease 2 (TMPRSS2) to enter the cells [27]. It is important to note that receptors for SARS-CoV-2 are also extensively expressed in the gastrointestinal tract, thus the alimentary system was also identified as an alternative transmission route of the virus [28,29,30].

The respiratory epithelium is equipped with cytosolic viral sensors and thus upon viral infection and active viral replication epithelial cells induce the production of antiviral IFNs, dominantly type III IFNs and to a lesser extent type I IFNs [31].

Type III IFNs differ from type IFNs in many aspects. Type I IFNs are produced by all nucleated cell types, and their receptor is ubiquitously expressed. In contrast, the main source of type III IFNs are epithelial cells but macrophages, monocytes and dendritic cells are also capable of producing them. The heterodimeric receptor of type III IFNs composed of IFN-λ receptor 1 (IFNLR1) and IL-10 receptor subunit-β (IL-10RB) is mainly expressed on epithelial cells and on some immune cells including pDCs, monocyte-derived DCs, monocytes, macrophages, neutrophils and B cells in humans that leads to more cell type specific immune responses. Despite the fact that type III IFNs use the same signaling pathway as type I IFNs [32,33], there are some differences in the antiviral responses induced by these cytokines. Type III IFN-mediated interferon-stimulated gene (ISG) expression is long-lasting, but lower in amplitude compared to the strong and rapid type I IFN induced ISG expression. Moreover, in contrast to the systemic type I IFN response, type III IFNs are less inflammatory and limit viral transmission and dissemination mainly at mucosal surfaces [32,33].

According to the newest data SARS-CoV-2 can easily overcome the first line of defense of the immune system provided by the airway epithelium. Due to the highly effective immune evasion strategies of SARS-CoV-2 the virus induced type I and type III IFN responses are delayed in the nasal epithelium compared to other respiratory viruses [34]. SARS-CoV-2 is able to counteract IFN signaling via several mechanisms. The virus can conceal pathogen-associated molecular patterns (PAMPs), disrupt the signaling cascade of IFN induction, suppress IFN action and interfere with IFN synthesis [35,36]. The emerging variants of SARS-CoV-2 also show increased resistance to IFNs that is the consequence of new non-structural protein (NSP) mutations of the virus [37].

If a virus passes through the first line of the defense, it has to face the innate immune response provided by macrophages and dendritic cells. Compared to the epithelium, these cells dominantly produce type I IFNs such as IFNα and IFNβ. Type I IFNs are more pleiotropic cytokines compared to type III IFNs. Besides inducing an antiviral state, type I IFNs also enhance antigen presentation, support natural killer (NK) cell functions, regulate B and T cell responses [33,38]. Furthermore, it is important to note that type I IFNs also control physiological processes such as the maintenance of synaptic plasticity of the central nervous system, the regulation of hematopoietic stem cell niche function, bone remodeling and microbiota-driven optimal antiviral responses [24]. The latest IFN-mediated process is extremely important in maintaining a baseline IFN production in the body that also supports an optimal IFN signature of the individuals [39,40,41].

The IFNα subtype is mainly produced by pDCs via activation of viral nucleic acid sensing endosomal Toll-like receptors (TLRs), while IFNβ is preferentially secreted by macrophages and myeloid DCs through stimulation of cytosolic retinoic acid-inducible gene (RIG)-I-like helicases (RLRs). Based on their unique capability to produce huge amounts of type I IFNs, pDCs are referred to as “professional” type I interferon-producing cells, which can produce 3–10 pg/cell of IFNα upon viral infection [42]. This is 10 to 100-fold more compared to the IFNα production of monocytes [43]. Besides type I IFNs pDCs are also capable of producing type III IFNs and are equipped with the type III IFN receptor as well [44]. Viruses and synthetic endosomal TLR agonist such as CpG-A can also induce type III IFN production in pDCs; however, compared to IFNα, pDCs produce IFNλ1 at levels approximately 10-fold lower. In general, a larger percentage of pDCs produce IFNα than IFNλ upon viral infection and IFNλ mostly serves as an autocrine signal that increases IFNα and IFNλ production of pDC and thus promotes pDC survival [44].

Investigating the IFN landscape in mild and severe COVID-19 cases it was revealed that IFNλ1 and IFNλ3 are dominant in the upper airways of mild COVID-19 patients and are responsible for the expression of protective ISGs. In contrast, in severe-to critical patients increased expression of type I IFNs and IFNλ2 can be observed. Similarly, in the lower airways of severe COVID-19 patients type I IFN and IFNλ2 expression is increased, whereas IFNλ1 and IFNλ3 expression is decreased that goes hand in hand with low ISG induction and high pro-apoptotic p53 expression [45]. These results indicate that spatio-temporal regulation of IFN responses is critical to overcome viral infections.

Based on previous studies on Middle East respiratory syndrome (MERS) and Severe Acute Respiratory Syndrome (SARS) infections and on the newest data derived from COVID-19 studies, the timing of the IFN response determines the course of the disease. When the viral load is still relatively low an early IFN induction results in rapid viral clearance and mild disease. In contrast, high viral load leads to delayed IFN response due to viral evasion mechanisms. Delayed IFN response promotes viral persistence and stimulates the pro-inflammatory cytokine production by innate immune cells resulting in overzealous inflammation and tissue damage [46,47]. Based on these data it can be concluded that the individual’s IFN signature fundamentally determines the severity of COVID-19. A recent review with an excellent summarizing table provided a thorough overview of literature data on the features of IFN response in COVID-19 patients [48].

## 3. The Role of pDCs in COVID-19

In the previous section we have emphasized the importance of optimal type I IFN response in COVID-19. As previously mentioned pDCs are the main producers of IFNα and according to the newest studies well-functioning pDCs are crucial to overcome SARS-CoV-2 infection.

In vitro studies showed that human pDCs are resistant to SARS-CoV-2 viruses due to the lack of ACE-2 and TMPRSS2 receptors, which are required for the cellular entry of SARS-CoV-2. However, instead of these proteins SARS-CoV-2 can use the transmembrane neuropilin 1 receptor (NRP1, also called blood dendritic cell antigen 4 [BDCA4]), the specific cell surface marker of pDCs, to enter the cells [49]. Previously it was observed that antibody ligation of BDCA4 impaired type I IFN production of pDCs [50,51], and lately it was also revealed that the binding of SARS-CoV-2 to BDCA4 on pDCs also decreased the type I IFN responses of pDCs that can serve as an evasion mechanism for the virus [26]. 

Another study also revealed that pDCs are refractory to SARS-CoV-2 infection under in vitro conditions. Upon SARS-CoV-2 stimulation, the signs of viral replication can not be observed in pDCs [52], and more interestingly SARS-CoV-2 stimulation increased the viability of pDCs compared to medium condition [52]. In addition, upon SARS-CoV-2 stimulation pDCs mount a robust type I IFN response by effectively producing type I IFNs. For instance, the concentration of IFNα2 can even reach 80 ng/mL [52]. Furthermore, the SARS-CoV-2-exposed pDCs can be divided into 3 activated subpopulations based on their cell surface co-stimulatory molecule expression, and this diversification can also be observed upon coculture with SARS-CoV-2 infected cells [52]. Upon SARS-CoV-2 stimulation pDCs mainly diversify into P1-pDCs (PD-L1+CD80−), which subset is characterized by high type I IFN production [49]. Infected cells also efficiently induce a P1 dominant diversification [53]. The type I IFN production of activated pDCs seems to be dependent on TLR7 signaling upon SARS-CoV-2 stimulation [49], and their diversification and cytokine production is related to the adaptor molecules IRAK4 and UNC93B1 [52]. The TLR7 activation-induced signaling also promotes the effective antiviral action of pDCs via the so-called interferogenic synapse in response to SARS-CoV-2 infected cells [54]. Overall, these in vitro studies indicate that pDCs can induce an effective type I IFN-dependent antiviral response against SARS-CoV-2 by inducing an antiviral state in the host cells to inhibit viral replication and by facilitating the antiviral actions of various innate and adaptive immune cells (Figure 1).

Besides the in vitro experiments, numerous in vivo studies proved that pDCs are crucial to mount an effective antiviral response against SARS-CoV-2. Several studies showed that pDC number is reduced in COVID-19 patients [55,56,57]. Furthermore, the decreased pDC number in COVID-19 patients negatively correlates with disease severity [57,58,59,60]. It was found that the immune landscape of patients differs depending on the severity of the disease. pDC frequency was lower in severe cases and that correlated well with disease severity. Moreover, clinical improvement of patients went hand in hand with increasing pDC frequency showing a dynamic process [61]. Another study also found that pDC frequency and number are decreased in asymptomatic patients compared to healthy donors and in hospitalized patients their level is dramatically reduced. Moreover, in asymptomatic patients, the high type I IFN producing P1-pDC population was dominant, while in hospitalized patients mainly the P2 subgroup (PD-L1+CD80+) with lower capacity to produce antiviral IFNs was observed [49,52]. Single-cell RNA sequencing showed that in the pDCs of severe COVID-19 patients the expression of pro-apoptotic molecules is increased, whereas their TLR7 and DHX36 expression are lost, and their antiviral effects and cytotoxic functions are decreased compared to pDCs from patients with moderate disease or cells from healthy controls [62]. The pDCs of hospitalized patients are characterized by decreased type I IFN and increased pro-inflammatory cytokine (TNF-α, IL-6) production compared to asymptomatic individuals [49].

Single-cell RNA sequencing of bronchoalveolar lavage fluid (BALF) from COVID-19 patients showed that in the BALF of severe/critical COVID-19 patients lower proportions of pDCs can be found compared to those with moderate infection [63]. Sánchez-Cerrillo et al. also found that pDCs are diminished from the blood of critical patients, and no pDCs were found in bronchoscopy infiltrates as well [64]. The decreased pDC number in the blood can be explained by the fact that pDCs migrate to the lungs upon inflammation; however, in critical cases, pDCs are also depleted in the lung. In case of extremely severe infection, it is due to the hyper-inflammatory landscape of the lung, which milieu impairs the viability and type I IFN producing capacity of pDCs.

In the context of chronic viral infections, high viral load induces pDC exhaustion, which means that pDCs tune down their type I IFN secretion and eventually die by apoptosis. This promotes viral replication and decreases the efficiency of innate immune responses [65]. In line with that, a study showed that pDCs from COVID-19 patients are characterized by decreased mTOR signaling and IFN production [58]. Moreover, pDCs displayed an apoptotic gene signature, which positively correlated with disease status and severity [57].

The type I IFN producing ability of pDCs is highly affected by the cytokine milieu of the inflamed lung and it is negatively regulated by pro-inflammatory mediators such as prostaglandin E2 (PGE2), IL-1β, IL-10 and TNF-α [66,67,68,69,70,71,72], which are highly elevated in COVID-19 patients [60,73].

It is also important to note that IL-3, which is mainly produced by T cells is an essential survival factor for pDCs and is also depleted in COVID-19 [74]. Benard et al. identified IL-3 as a prognostic marker for COVID-19 severity and outcome. Low IL-3 level correlates with increased viral load, mortality and severity. Non-survivors had lower T cell numbers and in COVID-19 patients the number of T cells correlates with pDC number in the plasma and BALF as well. In addition, in the BALF of COVID-19 patients with pulmonary manifestation a positive correlation was found between IL-3 and CXCL12 levels. The authors found that IL-3 derived from T cells induced the secretion of CXCL12 by epithelial cells and this chemokine mediated the recruitment of pDCs into the lung [75] (Figure 2).

In conclusion, these studies highlight that optimal pDC number and type I IFN response are vital to control SARS-CoV-2 infection and prevent the development of severe disease. Thus, all kind of diseases and conditions, which are characterized by low pDC frequency and decreased type I IFN production are risk factors for severe/critical COVID-19. The features of pDCs in COVID-19 are extensively reviewed in a recent paper [26], which thoroughly details the positive correlation between pDC function and COVID-19 severity, and provides a summary table about the observations regarding the fate of pDCs during COVID-19.

## 4. Risk Factors of COVID-19

While the mortality rate of COVID-19 is 0.9% in the healthy population, this ratio is significantly higher in patients with cardiovascular disease (10.5%), diabetes (7.3%) and hypertension (6%) [76]. The reason for the more severe disease course is that these conditions are all associated with chronic inflammation [18,77]. In diabetes, persistent inflammation due to hyperglycemia promotes easier entry of the virus into cells, and inhibits T-cell functions leading to a higher viral load. In addition, an exaggerated immune response predisposes to the development of a cytokine storm [78]. Excessive baseline activity of immune cells is also observed in hypertension, which reduces the efficiency of virus elimination and leads to more severe airway inflammation [79]. Heart disease is also associated with a poor prognosis of COVID-19, since infection associated fever and tachycardia increase the body’s need for oxygen that puts a heavy strain on a sick heart, and the virus can also damage the heart muscle directly or by inducing a cytokine storm indirectly [80].

In addition to the above mentioned diseases, which correlate with a more severe prognosis of COVID-19, there are a number of other risk factors that are closely linked to the body’s dysregulated type I IFN response or decreased type I IFN production due to various congenital or acquired causes [23,25]. The factors or conditions, which negatively affect the body’s type I IFN production result in a higher viral load due to inadequate virus elimination and thus lead to a more severe course of COVID-19. Such factors influencing type I IFN responses may include genetic defects of the antiviral immune response, or factors modulating baseline IFN signature, such as sex, age, and the condition of the microbiome, which promotes a steady-state basal IFN production. In addition, obesity, pregnancy, and viral infection-induced autoantibody production also result in an altered IFN response of the body. Furthermore, immunosuppression-associated chronic conditions that result from endogenous immunodysfunctions or immunosuppressive therapies are also risk factors for COVID-19. These conditions are also mostly associated with impaired functionality of pDCs, the main sources of type I IFNs in antiviral responses. In the following section, we detail those type I IFN response-associated risk factors, which may contribute to the development of more severe symptoms of COVID-19 (Figure 3).

## 5. Type I IFN-Associated Risk Factors in COVID-19

### 5.1. Genetic and Congenital Factors Associated with Reduced Antiviral IFN Production

It has long been known that defective variants at 13 genetic loci contribute to the development of influenza virus-induced severe pneumonia (*IRF7, IRF9* and *TLR3* genes), adverse events to live attenuated virus vaccines (*IFNAR1, IFNAR2, STAT2* genes) or herpes simplex encephalitis (*TLR3, UNC93B1, TICAM1, TRAF3, TBK1, IKBKG, IRF3, IFNAR1, STAT1* genes). These congenital gene defects impair the TLR3- and IRF7-dependent type I IFN responses [81,82]. In an international cohort, it has been shown that approximately 3.5% of patients with critical COVID-19 carry loss of function mutations at these loci [83]. In 23 people from the 659 patients with severe COVID-19 autosomal recessive (*IRF7, IFNAR1*) and autosomal dominant (*TLR3, UNC93B1, TICAM1, TBK1, IRF7, IRF3, IFNAR1, IFNAR2*) deficiencies were found. 10 patients were also characterized by low IFNα levels. It seems that the penetration values are higher for autosomal recessive mutations than for autosomal dominant gene defects [83,84]. *TLR3-, IRF7*-, or *IFNAR1*-deficient cells are highly sensitive to SARS-CoV-2, and *IRF7*-deficient pDCs are unable to produce type I IFNs upon viral exposure [83]. Furthermore, *IFNAR1*-deficient cells do not respond to type I IFN stimulation. The studies identified two patients (49 and 50 years old) with autosomal recessive *IRF7* deficiency and two other patients (26 and 38 years old) with *IFNAR1* mutation. Prior to COVID-19 pneumonia, none of the four patients required hospitalization for severe viral illness, highlighting that in contrast to seasonal influenza viruses these mutations have a higher penetrance for COVID-19 [84]. Moreover, van der Made et al. identified four young male patients, who carried the loss-of-function variants of *TLR7*. They all suffered from severe COVID-19 and were characterized by impaired type I and type II IFN production [85]. Another study found that in men under 60 years recessive *TLR7* deficiency accounts for 1% of critical COVID-19 cases [86]. In addition, the *interferon-induced transmembrane protein 3 (IFITM3)* gene encodes a protein, which is critical to restrict viral replication and to inhibit membrane fusion. A study found that homozygosity for the C allele of the rs12252 SNP in *IFITIM3* gene leads to more severe disease in an age-dependent manner [87]. This genetic variant is also associated with COVID-19 mortality in the Arab population [88].

Genome-level association studies (GWAS) have so far identified 4 chromosomal regions that may be associated with severe COVID-19. The first such region is located on chromosome three, and all that is currently known is that it encodes six genes (*SLC6A20, LZTFL1, CCR9, FYCO1, CXCR6,* and *XCR1*); however, their functions are still unexplored [89,90]. Furthermore, three other regions found in a recent GWAS analyzing 2244 critically ill patients from UK intensive care units were also identified in an international GWAS comparing hospitalized COVID-19 patients with other members of the population [90]. The odds ratio for heterozygous susceptibility alleles is between 1.2 and 1.4. Two of these three regions also contain genes involved in the antiviral immune response. The first region at chr12q24.13 includes the *OAS1, OAS2,* and *OAS3* genes and a group of ISGs required for RNase L enzyme activation. The second region, chr21q22.1, contains *IFNAR2,* which encodes the second chain of the IFN receptor [90].

We could expect that more alleles of the genes involved in the type I IFN response might be beneficial against COVID-19. However, according to the latest data it is not the case. People with Down syndrome have an extra chromosome, which encodes several gene involved in type I IFN response, for example the IFN receptor is also triplicated in trisomic cells. In the initial phase of the infection, the overactive type I IFN response might be advantageous, but later it fuels a detrimental inflammatory response due to the pleiotropic effects of type I IFNs [91].

Besides genetic defects, autoantibodies against type I IFNs are also associated with severe COVID-19. It was discovered that at least 10% of patients with critical COVID-19 pneumonia had autoantibodies that were able to neutralize large amounts of at least one, but typically more, type I IFN subtypes in vitro and in vivo. These IgG antibodies mainly neutralized IFNω, IFNα, or both, however, some patients had autoantibodies against all 13 IFNα subtypes. These autoantibodies were not found in any of the tested individuals with asymptomatic or mild SARS-CoV-2 and were present only in 0.33% of healthy individuals. Interestingly, these autoantibodies already existed in the patients prior to SARS-CoV-2 infection, and were the cause of severe disease rather than the consequence of the infection. The presence of these antibodies was associated with poor clinical outcome and increased mortality. It is noteworthy that 94% of patients with autoantibodies were male, half of them were over 65 years of age, and more than a third of them died from COVID-19. Overall, autoantibodies against type I IFNs are present in at least 3.5% of women and 12.5% of men with critical COVID-19 [92]. Another study showed that in vitro non-neutralizing anti-IFN antibodies were detected in 16% of patients, who were admitted to the intensive care unit (ICU) due to non-viral respiratory infection. However, neutralizing autoantibodies were only detectable in severe SARS-CoV-2 infected patients and their presence was associated with higher mortality and the development of multiple organ failure [93]. Wang and his colleagues also screened COVID-19 patients and healthy individuals for autoantibodies against extracellular and secreted proteins. They identified autoantibodies against type I IFNs in 5.2% of patients, who were hospitalized with COVID-19. Autoantibodies against type III IFNs (IFNλ2 and IFNλ3) were also found. Patients with type I IFN neutralizing antibodies were characterized by higher average viral load and extended durations of hospital admission [94]. It should be noted that only 2% of individuals with autoantibodies against type I IFNs produce autoantibodies against IFNβ [92]. However, autoantibodies are likely to be more common against the 13 IFNα subtypes and IFNω. Furthermore, genes encoding some of these IFN subtypes underwent strong negative selection, suggesting that they play an extremely important role in the antiviral response of the population [95]. Increased autoantibody production is probably due to an X chromosome-linked defect that is indicated by the increased involvement of men and the fact that one of the autoantibody-producing women suffered from a disease called incontinentia pigmenti, in which the inactivation of the X chromosome is skewed, and not random [92]. After the age of 65, autoantibody production is more likely because the composition of the immune system also changes with age. For instance, an atypical B-cell subpopulation might arise known as age-associated B cells (ABC), which differentiate into abnormal plasma cells characterized by increased autoantibody production [96]. In line with that, the incidence of neutralizing antibodies sharply rises after the age of 70. Neutralizing antibodies account for approximately 20% of both critical COVID-19 cases in the over 80s, and total lethal COVID-19 cases [97].

### 5.2. Influence of Biological Sex and Sex Hormones on Antiviral IFN Signature

Sexual dimorphism is observed not only in the physical appearance and behaviour of the sexes, but also in the context of autoimmunity and antiviral immunity [98,99,100]. It has long been known that women are less susceptible to viral infections than men due to their ability to develop a more effective antiviral response. Today, unfortunately, we can see how this observation is confirmed, as the currently raging COVID-19 pandemic affects men much more severely than women. For men, the mortality rate is 1.7 times higher in COVID-19, and sex differences are even more pronounced in the population over 30 years of age [101,102]. In a cohort longitudinal analysis of COVID-19 patients, higher level of IFNα was found in female patients [103]. The different type I IFN-producing capacity of pDCs in men and women plays a major role in this phenomenon. Genes involved in the antiviral response are often located on sex chromosomes or contain a hormone response element (HRE), so their expression is regulated by sex hormones and depends on the inactivation of sex chromosomes. 

X chromosome number affects the type I IFN response of pDCs. In a humanized mouse model, it has been shown that when CD34+ human stem cells isolated from women or men are transplanted into female or male mice, pDCs derived from female stem cells produce higher amounts of type I IFNs upon TLR7 stimulation than pDCs from male donor cells regardless of the sex of the recipient mice. These data suggest that the double X chromosomes in women provide immunological benefits as it may contribute to an enhanced immune response against infections [104]. A similar study examined the effect of X chromosome number and sex hormones on the TLR7-induced IFNα response of primary pDCs in healthy women, patients suffering from Turner syndrome, men, and transgender volunteers receiving hormone therapy. It has been found that the antiviral effect induced by type I IFNs is much more pronounced in healthy women than in men or women with Turner syndrome, where one of the X chromosomes is absent. However, the strength of the antiviral response did not correlate with serum sex hormone levels [105]. Furthermore, it is known that several genes encoded on the X chromosome involved in the TLR signaling pathway can avoid X chromosome inactivation and thus contribute to a stronger antiviral and humoral immune response. It has been observed that *TLR7* encoded on the X chromosome is biallelically expressed in the pDCs, B cells and monocytes not only of women (XX) but also of men with Klinefelter’s syndrome (XXY), and that immune cells with biallelic *TLR7* expression show greater transcriptional activity compared to monoallellic cells [106]. This was supported by another study showing that pDCs from women with biallelic *TLR7* expression are capable of greater IFN production than pDCs expressing only one *TLR7* allele [107]. These data may also explain why men with a single X chromosome have a higher mortality rate for COVID-19 compared to women [108,109]. PDCs in women with biallelic *TLR7* expression may produce a higher amount of type I IFNs and respond more rapidly to SARS-CoV-2 infection that may result in a better control of the infection in women [107]. These data are further supported by a recent study showing that loss-of-function mutation of *TLR7* on the X chromosome results in severe COVID-19 symptoms in young men that also indicates that the corresponding TLR7-mediated type I IFN response can play an essential role in overcoming the disease [85]. In addition, it is important to mention that pDC-derived type I IFNs regulate B cell activation and differentiation into plasma cells and are therefore essential to elicit an optimal antibody response against viral infections, therefore women may have an advantage over men in terms of antibody response regarding SARS-CoV-2 infection [102].

Sex hormones also affect antiviral immune responses. Due to their lipophilic nature, steroid hormones readily cross the plasma membranes of cells and, by binding to nuclear receptors, are able to affect the functions of immune cells, including pDCs [110]. Oestrogen is known to play an important role in regulating TLR-mediated immune responses in human and mouse pDCs. In mice, 17β-oestradiol (E2) has been reported to significantly increase CpG-B-induced IFNα production by spleen pDCs [111]. Consistent with this observation, E2 treatment in postmenopausal women also significantly increased TLR7/9 activation-induced IFNα production by primary pDCs. It has also been shown that E2 directly targets pDCs, as deletion of the oestrogen receptor α (ERα) in mouse pDCs suspended the positive effect of E2 treatment on TLR-induced IFNα induction [112]. In addition, impairment of oestrogen receptor signaling significantly reduces TLR7-induced IFNα expression in human pDCs from umbilical cord blood [104]. In another study, the ERα signaling pathway was found to induce increased IFNα secretion in TLR7-stimulated mouse pDCs through activation of the transcription factor IRF5, which is a positive regulator of the IFNα response of pDCs [113].

So far, only one study has examined the effect of androgens on the functions of pDCs. Dihydrotestosterone (DHT) has been shown to reduce TLR7-mediated IFNα production by pDCs isolated from the blood of healthy women. It was also found that pDCs in male infants produced less IFNα in response to TLR7 stimulation compared to female infants that can be explained by the early postnatal testosterone surge in 1–6-month-old male infants [114].

Based on the above data, it can be concluded that while oestrogens positively regulate the type I IFN response of pDCs, testosterone may negatively affect these processes. Thus, gender differences can greatly determine the strength of an individual’s immune response to viral infections as well as the efficacy of vaccines [115].

Observations to date have shown that COVID-19 is also more dangerous for pregnant women, which might be explained by the effects of progesterones. Pregnant women are less likely to have typical symptoms of SARS-CoV-2 infection such as fever, dyspnoea, and muscle pain, but are more likely to be admitted to the ICU or require invasive ventilation than other non-pregnant women of childbearing age. Of course, other risk factors for COVID-19, such as pre-existing comorbidities, ethnicity, chronic hypertension, pre-existing diabetes, high maternal age, and high body mass index (BMI), also carry the potential for more severe viral infections during pregnancy. Pregnant women with COVID-19 are at increased risk of preterm birth, gestational toxaemia, caesarean section, maternal mortality, and admission to the ICU. Newborns are also more likely to require neonatal intensive care [21,116].

During pregnancy, a number of physiological changes occur in the body, including changes in the function of the immune system. It is known that in pregnant women, from the start of implantation, the immune response shifts towards a Th2 type tolerogenic immune response that provides the optimal microenvironment for the development of the foetus in the maternal uterus. The predominant Th2 immunity then switches to a Th1 dominance at the end of pregnancy that is required for labour induction [117]. Along with the number of circulating NK cells, the number of pDCs also decreases as pregnancy progresses [118,119]. Furthermore, in vitro experiments have already showed that after H1N1 infection, pDCs of pregnant women produce less IFNα compared to non-pregnant women [119]. This may explain why pregnant women are more severely affected during influenza as well as COVID-19 pandemics [120]. Progesterone hormone levels also increase in women during pregnancy, and their immunosuppressive properties and negative effects on the functions of pDCs are well known [121]. In contrast to oestrogen, progesterone and its synthetic analogues inhibit the activity of innate immune cells and negatively regulate the secretion of type I IFNs in human pDCs [122]. In vitro experiments have shown that progesterone and depo-medroxyprogesterone acetate (DMPA), a synthetic form of progesterone, inhibit TLR9 activation-induced IFNα secretion in mouse and human pDCs. In vivo vesicular stomatitis virus (VSV) infection has also been shown to significantly lower serum IFNα levels in DMPA-treated mice compared to DMPA-untreated mice. The inhibitory effect of progesterone may be due to the inhibition of TLR9 activation-induced nuclear translocation of the transcription factor IRF7 in pDCs [123,124]. These data indicate that the enhanced tolerogenic responses to protect the foetus during pregnancy and the negative effect of progesterone on pDCs’ type I IFN production make pregnant women more vulnerable to viral infections including SARS-CoV-2 infection.

The question may arise whether it is safe to use type I IFN therapy in pregnant women. A meta-analysis concluded that IFNα did not significantly increase the risk of developmental abnormalities, miscarriages, stillbirths, or preterm births in women exposed to IFNs during pregnancy [125]. Thus, in pregnant women suffering from severe COVID-19, if the possibility of IFN therapy arises, it may be safe to use.

### 5.3. The Role of Age in Impaired IFN Production

Age is a very prominent clinical risk factor of COVID-19 mortality [126]. This is supported by the fact that the mortality rate of COVID-19 was found to be lower in patients under 60 years of age (1.4%) than in those over 60 years of age (4.5%) [127]. Increased morbidity and mortality in the elderly are likely to be caused by a shift in the innate immune system towards inflammation, as well as age-related cellular changes and abnormalities in antiviral signaling pathways leading to delayed, prolonged type I IFN production. In the elderly, the basic inflammatory phenotype may result in a late type I IFN response during viral infections that has been previously observed in the case of SARS-CoV infection as well [128]. Delayed antiviral type I IFN response leads to increased tissue damage and cytokine storm, which is also the characteristic of severe COVID-19 [128,129]. With regard to SARS-CoV infection, it has also been previously reported that the frequency of pro-inflammatory macrophages and alveolar macrophages in the lung may also shift due to the disruption of IFN production [130]. Furthermore, during viral infections, type I IFNs support NK cell activation, while inhibit pathological responses mediated by neutrophil granulocytes and type II innate lymphoid cells (ILC2) in the infected mucosa [129,131,132].

In addition, the efficiency of the early type I IFN response is decreased with age due to the descending number of IFN-producing macrophages and DCs, and impairment of signaling pathways implicated in IFN production [133,134,135,136]. While the myeloid DC population persists with advancing age, a decline in the number and function of pDCs was reported in association with aging [135,136,137,138,139,140]. In the elderly, the decline in IFN-producing ability of pDCs is partly due to the decreased TLR7/9 expression [135] and functional impairment of IRF7 [138]. These processes are associated with increased reactive oxygen species (ROS) levels and cell damage observed in aging cells [141]. Furthermore, aging also affects the RIG-I/MDA-5 signaling pathway, as proteasomal degradation of TRAF3 is increased in elderly human monocytes, making IRF3 activation less efficient and thus results in lower production of antiviral IFNs [133]. In contrast to adults, the nasal epithelial cells, macrophages and DCs of children are abundant in receptors such as RIG-I and MDA5. The high baseline expression of these sensors results in a stronger, immediate antiviral response against SARS-CoV-2 that can partially explain the lower sensitivity of children to the more severe symptoms of COVID-19 [142].

Thus, impairment of type I IFN production pathways, delayed IFN response, and pDC dysfunction in elderly individuals greatly reduce the chances of overcoming SARS-CoV-2 infection [129,143].

### 5.4. The Role of Microbiome in Antiviral IFN Production

A healthy gut microbiome is essential to support the host’s immune responses. On the one hand, it prevents the activation of pro-inflammatory cascades, on the other hand, it prepares the body for future viral infections [144,145]. However, in the state of dysbiosis, these protective functions are impaired. Many studies suggest that the clinical manifestation and severity of COVID-19 may be linked to gut dysbiosis [144,145,146,147]. Furthermore, SARS-CoV-2 infection can also alter the microbial composition of the lung indicating that serious inflammation occurs in lung tissues. The level of inflammation detected in the lung was significantly correlated with the levels of pathogenic microorganisms and SARS-CoV-2 [148]. Dysbiosis of the respiratory tract in hospitalized COVID-19 patients leads to accelerated destabilization over time and correlates with disease severity and systemic immune activation [149]. In intubated patients an enrichment of *Staphylococcus* species can be observed. Moreover, the small commensal DNA viruses, *Anelloviridae* and *Redondoviridae* showed increased titer and colonization in severe COVID-19 as well [149]. In the upper respiratory tract the bacterial load, bacterial richness consistently increased, while the abundance of an amplicon sequence variant, *Corynebacterium_unclassified.ASV0002* decreased, as disease severity increased [150].

Various commensal intestinal bacteria with beneficial immunomodulatory potential, such as *Faecalibacterium prausnitzii, Eubacterium rectale*, and bifidobacteria were reduced in COVID-19 patients and their frequency remained low even after 30 days of recovery from COVID-19. The decline in beneficial gut bacteria was correlated with increased disease severity, and elevated levels of inflammatory markers and cytokines in the patients ’plasma. This may suggest that the microbial disturbance that persists after disease resolution may contribute to post-COVID syndrome [151]. Another study also found that decreased commensal species and increased opportunistic pathogenic species characterize the gut of COVID-19 patients. Severe illness was associated with the abundance of *Burkholderia contaminans*, *Bacteroides nordii* and *Blautia* sp. *CAG 257*. The abundance of *Burkholderia contaminans* was correlated with higher levels of inflammation and lower number of immune cells [152]. In addition, a decrease in *Lactobacillus* and *Bifidobacterium* species, which play important roles in protecting against intestinal infections by stimulating intestinal functions, promoting immune responses, and preventing the overgrowth of pathogenic species, has been observed in COVID-19 patients with intestinal dysbiosis [153].

Since the commensal microbial flora is vital to maintain the baseline IFN secretion in the human body, dysbiosis might lead to a decreased antiviral immune response. The stimulatory signals from commensal bacteria keep the immune cells as well as the stromal cells in constant state of antiviral readiness. Among others, they maintain the constitutive, low-level IFN production of pDCs [154], the baseline activity of mononuclear phagocytes and NK cells [155], the baseline production of IFN by lung stromal cells, and thus the constitutive expression of antiviral Mx proteins [40]. It is important to note that antibiotic therapy can easily destroy this vulnerable system, since antibiotics not only target pathogenic bacteria but also kill or drastically reduce the numbers of commensal bacteria, which are responsible for sustaining tonic levels of IFN signals, and thus antibiotics eliminate the body’s baseline antiviral state [40,41], and increase the risk of viral infections and inflammatory conditions [39]. This phenomenon was elegantly demonstrated using a mouse model. When mice with healthy intestinal flora were infected with influenza virus, 80% of the mice survived. However, with antibiotic pre-treatment, only one-third of the mice survived the infection, but faecal transplantation could rescue mice from pathogen-induced death/sepsis. These results indicate that a healthy intestinal flora provide a strong protection against influenza, as the gut microbiota-driven systemic antiviral immunity was already active when the virus entered the body. On the contrary, in the absence of intestinal bacteria, the antiviral genes only turn on when the immune response is triggered. However, this sometimes happens too late, when the virus has already multiplied in the body and thus the high viral load leads to an exaggerated, detrimental immune response [40]. In line with that, a significant correlation was found between previous antibiotic exposure and increased severity of COVID-19 in Spain [156]. Thus, it can be assumed that among many other factors, dysbiosis caused by the overuse of antibiotics, may be listed as a risk factor for severe COVID-19.

These data suggest that the use of probiotics as a prophylaxis may be advisable to reduce the incidence of respiratory infections [157,158]. Several data indicate that prebiotics and probiotics are able to enhance the type I IFN response of pDCs through TLR9 stimulation and thus provide a more effective antiviral response [159,160,161,162]. Besides probiotics, it may be advisable to increase the intake of anti-inflammatory foods, such as vegetables and fruits, as a high-fiber diet serves as a good source of carbohydrates for beneficial bacteria. In addition, foods with a high polyphenol content, such as vegetables, fruits, cereals, tea, coffee, dark chocolate or cocoa powder, have prebiotic or antimicrobial properties and thus can effectively inhibit the replication of pathogens in the body [163,164,165]. Therefore, a proper, personalized diet might help to prevent coronavirus infection and might contribute to patients’ recovery, as well as might help to eliminate dysbiosis caused by the infection and restore the gut microbiota after recovery from COVID-19.

### 5.5. Obesity and Antiviral IFNs

So far it seems, that obesity also predisposes to a more severe course of SARS-CoV-2 infection [166]. Diabetes, hypertension, and cardiovascular diseases, which are risk factors of COVID-19, are commonly associated with obesity. For example, one study found that 74% of diabetics were obese, which may further exacerbate the severity of COVID-19 in this disease group [167]. When obesity, diabetes, hypertension, and dyslipidemia occur together, it is called metabolic syndrome, a disease, which is also associated with increased COVID-19 mortality [168]. Thus, obesity is not only a single risk factor, but by acting synergistically with other underlying diseases it may further increase the incidence of critical SARS-CoV-2 infection.

According to a comprehensive study examining data from 5700 hospitalized patients infected with SARS-CoV-2, obesity (41.7%) is the second most common comorbidity in COVID-19 after hypertension (56.6%) [169]. According to a French study, 47.6% of patients in the ICU had a BMI above 30 kg/m^2^, while 28.2% had a BMI above 35 kg/m^2^ [170]. Reports from two Spanish ICUs also confirmed that obesity is the most common comorbidity that occurred in half of the patients admitted to hospital [171]. However, data from 6 New York University hospitals show an inverse correlation between BMI and age among those admitted to the ICU. Although the risk of severe disease in SARS-CoV-2 infection increases with age, younger patients with critical disease were more likely to be obese [172]. A meta-analysis found that obese people were 113% more likely to be hospitalized, 74% more likely to be admitted to an ICU, and 48% more likely to die [173].

More severe COVID-19 symptoms in obese individuals may be caused by a weaker and prolonged type I IFN response that results in a decreased antiviral immune response. In obese people, the serum level of the hormone leptin produced by fat cells is high, which may indicate leptin resistance. Leptin may induce the expression and activation of suppressor of cytokine signaling (SOCS) 3 and while decrease the type I IFN response in obese individuals [174,175,176]. Type I IFNs and leptin use the same JAK–STAT signaling pathway that can be inhibited by SOCS3 and that results in a lower IFN response to viral infections in obese individuals [174,177]. It has recently been shown that the baseline SOCS3 expression is increased and correlates with a decreased type I IFN response in obese patients [177]. Due to this reason, obese individuals are also much more susceptible to infections and are characterized by higher mortality during seasonal influenza epidemics [178,179]. In addition, increased inflammatory cytokines levels, enhanced M1 polarization of lung macrophages and impaired IFN response and ISG induction by respiratory epithelial cells and macrophages can be observed in obesity that can eventually lead to more severe pneumonia and lung damage in obese individuals [178,179]. Furthermore, the diet of obese individuals is generally high in fat, which can also lead to dysbiosis, thereby further reducing the intensity of the type I IFN response [174]. Collectively, obese patients are characterized by reduced IFN production, and thus might provide a microenvironment that allows the emergence of novel virulent variants of the virus [178].

### 5.6. Underlying Chronic Medical Conditions Associated with Impaired IFN Response Causing Immunosuppression

We had already mentioned that chronic diseases such as diabetes, hypertension, obesity are listed as the underlying cause in the majority of COVID-19 mortality. However, the proportion of another group of diseases, the immunosuppression-associated chronic diseases, which are caused by either endogenous immunodysfunctions or immunosuppressive treatments, is also remarkably high [180]. This group includes, but is not limited to, primary and secondary immune deficiencies, cancers, chronic renal failure, post-transplant organ status, and autoimmune diseases. Particular attention should be paid to this group of diseases, as not only the patients themselves are at increased risk, but also their immediate environment, as immunosuppressed individuals can serve as “reservoirs” for viruses and might remain infectious for up to several months [181,182,183]. Furthermore, it may also be of concern that viral pneumonia may occur atypically with low inflammatory markers in these patients, but later can be associated with a more severe disease course [184].

Studies have shown that patients with primary and secondary immunodeficiencies are characterized by increased morbidity and mortality from COVID-19 compared to the general population [185]. A meta-analysis also supports increased mortality in patients with chronic renal failure associated with immunosuppression [186,187]. Cancer patients with COVID-19 are 3.5 times more likely to be admitted to the ICU and to need mechanical ventilation, and are more prone to infections with SARS-CoV-2, which is eliminated later from their bodies compared to the general population [188,189]. In the case of autoimmune diseases, however, the situation is more complicated. Although COVID-19 is more severe compared to influenza in autoimmune patients [190], it appears that low-dose immunosuppressive therapy may provide protection against the complications of COVID-19 in these patients [191,192].

Immunosuppressive agents used to treat certain autoimmune conditions also affect the production of type I IFNs, as well as the functions of pDCs, and may predispose to more severe viral infections. For example, steroids have been reported to reduce the number and type I IFN responses of pDCs in systemic lupus erythematosus (SLE) patients; however, it is important to note that after the discontinuation of glucocorticoids, both pDC number and IFNα levels recovered rapidly in the patients [193,194]. Hydrochloroquine also reduces type I IFN production of TLR7 or TLR9 activated pDCs in patients with SLE [195] and also inhibits TLR9 activation-induced type I IFN production by pDCs of cutaneous lupus erythematosus patients [196]. Furthermore, the active form of mycophenolate mofetil, mycophenolic acid, is also able to dose-dependently reduce CpG-induced type I IFN secretion in pDCs of SLE patients by inhibiting nuclear translocation of IRF7 [197]. Furthermore, baricitinib, which inhibits the JAK/STAT pathway, is able to inhibit IFN secretion by pDCs and thus increases the risk of varicella reactivation as well [198,199].

It is important to note that besides the above mentioned immunomodulatory effects, some IFN response inhibitors, also exhibit direct antiviral activity as well. For example, chloroquine interferes with different stages of the viral life cycle including viral entry, uncoating, assembly and budding. Via increasing endosomal pH chloroquine blocks virus-endosome fusion and is also able to inhibit posttranslational modifications of viral proteins by interfering with proteolytic processes [200,201]. Furthermore, mycophenolate mofetil was able to inhibit SARS-COV-2 replication in vitro. Similar antiviral activity was observed for calcineurin and mTOR inhibitors as well as thiopurine analogs against SARS-CoV and MERS-CoV strains [202]. Nevertheless, immunosuppressive agents may still be detrimental in the initial phase of COVID-19, since the weakened immune system cannot adequately control viral replication. However, in the later stages of the disease, the immunosuppressive effects of these drugs may be particularly beneficial, as they may prevent an overzealous immune response, the development of cytokine storm, and multi-organ failure. Thus, as previously mentioned, low-dose immunosuppression may have a beneficial effect in autoimmune patients, as it may alleviate the severe symptoms of COVID-19 caused by the body’s overactivated immune response [191,192].

## 6. Discussion

The severity of viral infections can greatly vary among individuals, as a wide array of endogenous and exogenous factors can affect an individual’s type I IFN response, which is one of the most important weapons of our immune system that can rapidly inhibit the replication of viruses [24]. Our body prepares to defend against viral infections long before viral exposure, owing to the constituent baseline type I IFN production by various tissues and cell types that creates a general antiviral state in the host [24]. In accordance with that, it was observed that the incidence and mortality of severe COVID-19 caused by SARS-CoV-2 is also significantly higher in individuals with an inadequate type I IFN response [203,204,205].

Interestingly, a robust type I IFN response can be observed in some patients that may result in early control of the infection and thus in a mild course. The sign of an effective IFN response in these patients may be reflected by the appearance of a skin condition called “COVID toes” [206]. The lesion is reminiscent of perniosis, which is an inflammatory condition caused by cold, and associated with red-purple discoloration and blistering of the acral areas. Histologically, edema of the epidermis as well as perivascular and perieccrine lymphocytic infiltration are present, and even microthrombi may form in the blood vessels. Similar changes can be observed in a rare cutaneous form of lupus, the so-called familiar chilblain lupus (FCL), which can be classified as an interferonopathy and is associated with increased type I IFN production [207]. Therefore, one might suppose that COVID toes are probably caused by the increased systemic type I IFN secretion and thus might serve as a marker for patients with efficient viral clearance and mild course of COVID-19 [206].

An individual’s IFN signature may also be adversely affected by viral evasion mechanisms as well as by host-dependent factors, which result in low or delayed IFN response and high viral load that impacts the severity of COVID-19 symptoms [203,208]. Thus, individuals with reduced IFN signature, such as patients with congenital defects of the IFN pathway, men, the elderly, people suffering from dysbiosis, obese or immunosuppressed individuals and pregnant women may benefit from type I IFN therapy in the early phase of the disease or as prophylaxis. Thus, the therapeutic application of type I IFNs in COVID-19 is in the focus of several ongoing clinical studies [209]. However, in order to treat COVID-19 more effectively, the type, dose, route and frequency of administration of the most optimal therapeutic IFNs subtype and the time of intervention needs to be optimized. PEGylated forms may be preferred over unmodified IFNs, since those can be administered by subcutaneous injection once a week. It should be noted that IFN injection elicits a systemic response and induces antiviral, pro-inflammatory and anti-inflammatory effects simultaneously. In contrast, the local action of inhaled type I IFNs in the airways may compensate for the production of IFNβ by epithelial cells and may have excellent preventive potential when used as a nasal drop [210,211,212,213,214]. Another important issue is the subtype of IFN used in therapy, since while IFNα has a strong antiviral effect, IFNβ also has immunomodulatory and antiproliferative effects as well [215,216]. It is also important to emphasize that the use of IFNs in patients with severe or critical COVID-19 is not advised, since IFNs significantly enhance the inflammatory state in the later stages of the infection and exacerbate cytokine storm and lung injury similar to the inflammation boosting effect of delayed type I IFN responses in severe COVID-19 patients [217].

The results of clinical trials regarding IFN therapies are encouraging [209]; however, prior to use it is essential to consider the condition of the patient, the effects of the treatment, and the possible side effects of the IFN subtype applied in these therapies.

## 7. Conclusions

In summary, an individual’s IFN signature is a major factor influencing the severity of COVID-19 outcome, which is strongly associated with the activity and number of pDCs. If lifestyle-related factors, which are detrimental to type I IFN responses or normal pDC functions, accumulate in a given population, a more severe epidemic can be expected.

The COVID-19 pandemic shed light on the importance of adequate/sufficient IFN responses and on the benefits of early or prophylactic IFN therapies. Therefore, a comprehensive analysis of the complex and pleiotropic effects of type I IFNs are needed to gain a clear understanding on the specific functions, kinetic profiles, tissue- and cell-type specific effects of all subtypes of type I IFNs (including the 13 subtypes of IFN-α along with IFN-β, IFN-ε, IFN-κ, IFN-ω, IFN-δ, IFN-ζ, and IFN-τ) and their efficacy against novel virus variants. In addition to SARS-CoV-2 infection, data obtained from such studies might also apply to other viral infections. Thus, patient with known risk factors might be targeted by the most optimal type I IFN therapy in the early stage of the infection that could significantly reduce disease severity and mortality upon the possible emergence of a new pandemic.

## Figures and Tables

**Figure 1 ijms-23-10968-f001:**
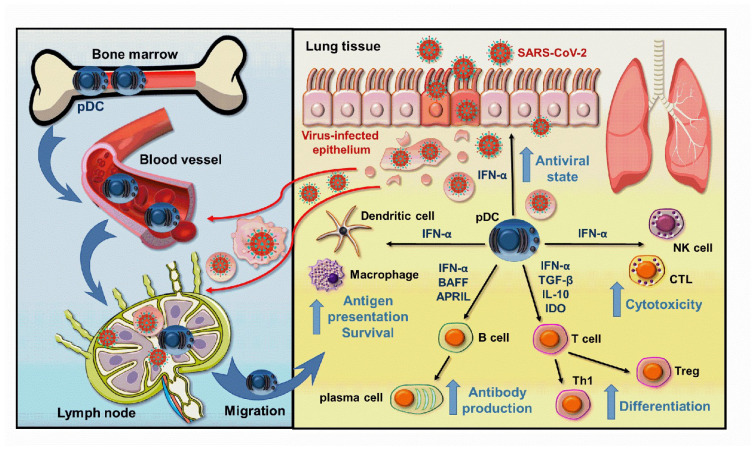
Beneficial effects of pDC-derived type I IFNs help to overcome SARS-CoV-2 infection. When the SARS-CoV-2 viruses break through the first line of defense ensured by the epithelial cells, viral particles or cell debris derived from virus-infected cells are delivered to the draining lymph nodes, where pDCs are stimulated to migrate to the entry site of the viruses. Here, the pDC-derived type I IFNs initiate an antiviral state in the host cells, which effectively blocks viral replication, and also promotes the activation and function of both innate and adaptive immune cells thereby creating an effective antiviral response. *APRIL: A proliferation-inducing ligand; BAFF: B-cell activating factor; IL: interleukin; IDO: Indoleamine 2, 3-dioxygenase; IFN: interferon; NK: natural killer; pDC: plasmacytoid dendritic cell; TGF: Transforming Growth Factor; Th: T helper; Treg: T regulatory; CTL: cytotoxic T cell*.

**Figure 2 ijms-23-10968-f002:**
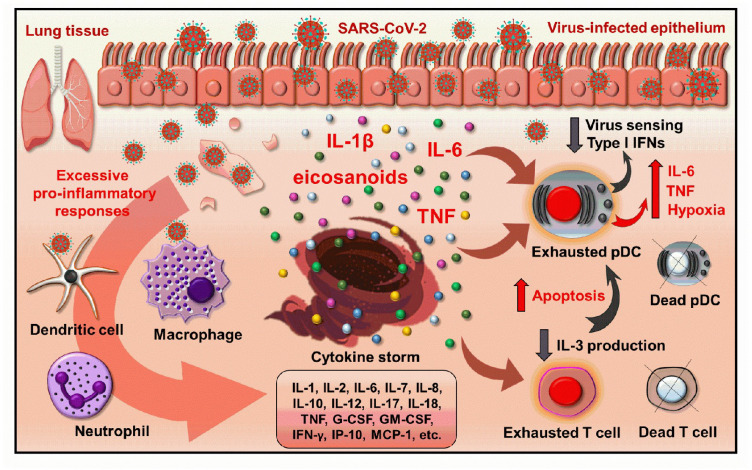
The inflammatory microenvironment in severe COVID-19 leads to the exhaustion and decreased antiviral potential of pDCs. The overactivation of immune cells in severe COVID-19 leads to an excessive production of pro-inflammatory cytokines and eventually cascades into a cytokine storm. This inflammatory environment drives pDC exhaustion that is characterized by high hypoxia and functional abnormalities of pDCs. However, the inflammatory environment has an inhibitory effect on the virus sensing ability and type I IFN production of pDCs, and their pro-inflammatory cytokine production comes to the fore that can further fuel the detrimental inflammatory circuit. The excessive inflammatory milieu also causes T cell exhaustion and leads to increased T cell death. Consequently, the levels of T cell-derived IL-3, which is an essential survival factor for pDCs, also drop and that leads to pDCs apoptosis. Thus, severe COVID-19 is associated with a reduced number of pDCs. *G-CSF: granulocyte colony stimulating factor; GM-CSF: granulocyte-macrophage colony stimulating factor; IFN: interferon; IP: interferon gamma-induced protein, MCP: monocyte chemoattractant protein; pDC: plasmacytoid dendritic cell; TNF: tumor necrosis factor*.

**Figure 3 ijms-23-10968-f003:**
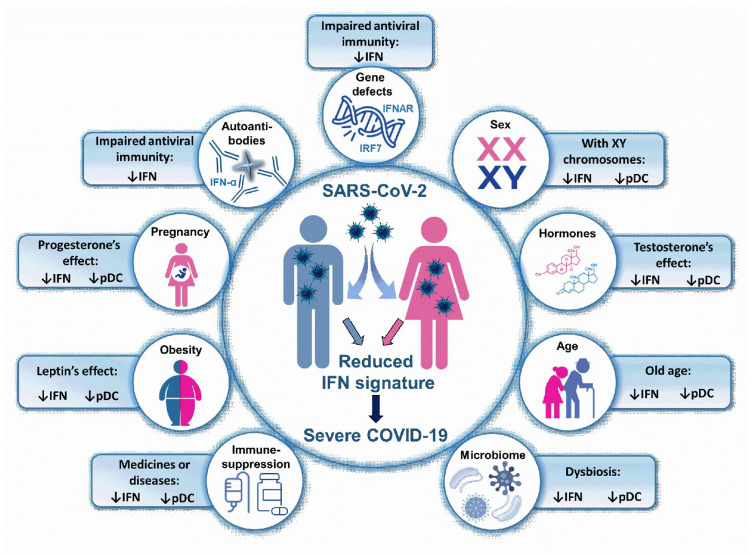
Type I IFN related risk factors in COVID-19. A low IFN signature predicts a more severe COVID-19 outcome. Thus, those conditions, which are associated with impaired type I IFN response or decreased pDC number may be risk factors for severe COVID-19. *IFN: interferon; pDC: plasmacytoid dendritic cell, IRF: interferon regulatory factor; IFNAR: interferon-α/β receptor*.

## Data Availability

Not applicable.

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
