# Peer review of "Correlation between Type I Interferon Associated Factors and COVID-19 Severity"

_ijms, 2022, doi:10.3390/ijms231810968_

Round 1

Reviewer 1 Report

The present review by Bencze et al describes the role of type I IFNs and subsequently the role of plasmacytoid DCs, the main producers of type I IFNs against SARS-CoV-2, the aetiological agent of COVID-19. Overall, this review is well organized providing all the necessary information for the readers to understand the innate immune response with specially emphasis to type I IFNs against SARS-CoV-2 and the effect of various factors to its production, such as sex, age, underlying diseases etc. The references are up to date. 

There are only minor comments. Specifically:

1. Figure legend of figure 2 is not discriminated from the main text. Authors should make the appropriate changes.

2. Line 503: Authors should add the reference.

Author Response

 Response to Reviewer 1.:

We are grateful for the positive evaluation of our manuscript and we thank to the Reviewer for the time and effort taken to evaluate our manuscript. Based on the Reviewer’s comments we made changes to the manuscript accordingly. Please see our responses below.

„1. Figure legend of figure 2 is not discriminated from the main text. Authors should make the appropriate changes.”

We thank for this perception of the Reviewer. We made the appropriate changes.

„2. Line 503: Authors should add the reference.”

The Reviewer is absolutely right and we thank the Reviewer for calling our attention to this mistake. In the revised manuscript we inserted the missing reference.

We would like to thank again the Reviewer for the careful reading of the manuscript that helped us to further improve the quality of our paper.

Reviewer 2 Report

This manuscript summarized the relationship between Covid-19 and interferon, specially produced by pDC. This manuscript should be further strengthened by addressing a concern as follow:

minor: please add all information of reference 153 as Journal of Diabetes.2021;13:420–429.

Author Response

Response to Reviewer 2.:

We are very grateful to the Reviewer for the time and effort taken to evaluate our manuscript. Based on the Reviewer’s suggestions we made changes to the manuscript accordingly.

„This manuscript summarized the relationship between Covid-19 and interferon, specially produced by pDC. This manuscript should be further strengthened by addressing a concern as follow:

minor: please add all information of reference 153 as Journal of Diabetes.2021;13:420–429.”

We thank the Reviewer for calling our attention to this mistake. In the revised manuscript we inserted the missing parts of the marked reference (new reference number 169).

We would like to thank again the Reviewer for reading of the manuscript and for helping us to further improve the quality of our paper.

Reviewer 3 Report

Some comments needed to adress in order to further process. Details of comments are  given below. 

Comments Details:

Line 33-35:  References are missing. Please update it. 

Line 49: Cytokine profile? What kind of cytokines are involved?

Line 72-77: Please start with new paragraph

Line 79: Infects to what organ? Is its attack can occurs to liver cells?

Line 86: replace can be produced with are produced by 

Line 149: replace INF signnature with INF biomarker

Line 165: also showed needed replacement with revealed 

Line 244: Please improve the figure caption

Line 451: Androgens role in viral replication activate or reactivate?

Line 673: Immunosuppressive agents can inhibit abnormal viral replication?. Please explain its roles. 

Author Response

Response to Reviewer 3.:

We are grateful for the positive evaluation of our manuscript and we thank to the Reviewer for the time and effort taken to evaluate our manuscript. Based on the Reviewer’s comments we made changes to the manuscript accordingly. Please see our detailed response below.

„Line 33-35:  References are missing. Please update it. „

The Reviewer is absolutely right and we thank the Reviewer for calling our attention to this mistake. In the revised version of the manuscript we inserted the missing reference.

„Line 49: Cytokine profile? What kind of cytokines are involved?”

We thank for the Reviewer’s question. We defined the cytokines, which participate in the symptoms of severe COVID-19 patients by causing excessive tissue inflammation. You can find the completed sentence in the revised version of the manuscript as follows:

 “In patients with this advanced stage of the disease, a cytokine profile resembling secondary haemophagocytic lymphohistiocytosis may be observed, which is characterized by elevated levels of IL-2, IL-7, granulocyte colony stimulating factor (G-CSF), IFN-γ inducible protein 10 (IP-10), monocyte chemoattractant protein 1 (MCP-1), macrophage inflammatory protein-1 α (MIP-1α), and tumor necrosis factor-α (TNF-α).”

„Line 72-77: Please start with new paragraph”

We appreciate the Reviewer suggestions, based on which we moved the indicated sentences in a new paragraph in the revised manuscript.

„Line 79: Infects to what organ? Is its attack can occurs to liver cells?”

We thank for the Reviewer’s question. In the marked sentence we wrote about the main entry route of the SARS-COV-2 to the human body, which is mainly the respiratory tract. The liver can also be affected due to the possible role of the gastrointestinal system in the transmission of the virus. However, the airway or gut epithelium are in the first line of the defense, which provide the first physical barrier between the pathogen-rich environment and the human body. We thanks the Reviewer to call our attention to the missing information that the gastrointestinal tract can also be a possible entry site for the SARS-CoV-2. Thus, we added this information to the text as follows:

“It is important to note that receptors for SARS-CoV-2 entry are also extensively expressed in the gastrointestinal tract, thus the alimentary system was also identified as an alternative transmission route of the virus (doi.org/10.1038/s41575-021-00416-6, 10.12998/wjcc.v9.i20.5427, 10.1111/ijcp.13893)”

Reviewer 4 asked us to provide additional data about the extrapulmonary manifestions of COVID-19 in the Introduction section, where we also listed the involvement of the liver by hepatocellular injuries indicating that hepatocytes can also be targeted by the virus, and we also added references related to SARS-CoV-2 liver tropism (References 12 and 13). You can find the appropriate sentences in lines 58-65 as follows:

“Besides the well-known respiratory pathology, various extrapulmonary manifestations of COVID-19 have already been reported highlighting the involvement of cardiovascular, genitourinary, gastrointestinal and central nervous system as well as the skin (10.1038/s41591-020-0968-3). The multi-organ involvement can be manifested by various symptoms including thrombotic complications, myocardial dysfunction, arrhythmia, acute coronary syndromes, acute kidney injury, gastrointestinal symptoms, hepatocellular injury, hyperglycemia and ketosis, neurologic illnesses, ocular symptoms, dermatologic complications, preeclampsia and fertility problems (10.1038/s41591-020-0968-3, 10.1093/brain/awab421, 10.1111/apm.13210, 10.1097/HJH.0000000000003213, 10.1038/s42255-022-00554-4, 10.1016/j.jhep.2020.05.002).”

However, if the Reviewer would allow us, we do not wish to detail the infection of the liver in the section about “The role of antiviral IFNs in COVID-19”, since this is not the main point of the section.

„Line 86: replace can be produced with are produced by” 

Based on the Reviewer’s comment, we rewrote the marked sentence in the revised version of the manuscript.

„Line 149: replace INF signnature with INF biomarker”

We thank for the Reviewer’s comment but we respectfully ask the Reviewer to allow us to keep the term “IFN signature” in the marked sentence. We think that there is some misunderstanding here. The term referrers to the increased expression of type I interferon regulated genes that is an important feature or clinical characteristic of an individual’s state. The term “biomarker” has a different meaning and does not fit to the content of the marked sentence.

„Line 165: also showed needed replacement with revealed” 

Based on the Reviewer’s comment, we corrected the sentence in the revised version of the manuscript.

„Line 244: Please improve the figure caption”

We appreciate the Reviewer constructive suggestion, based on which we rephrased the caption of Figure 2 as follows: “The inflammatory microenvironment in severe COVID-19 leads to the exhaustion and decreased antiviral potential of pDCs.”

„Line 451: Androgens role in viral replication activate or reactivate?”

We thank for this interesting question of the Reviewer. Several data from experiments with mice are available in the literature about the effects of androgens on viral replication. However, the results are controversial and do not reveal the direct effects of androgens on the viral replication. In some paper it was published that androgens enhance the replication and dissemination of the viruses (10.1128/jvi.01232-22, 10.1128/JVI.01763-09, 10.1128/JVI.06707-11). In contrast, although androgens influence the outcome of virus infection, do not alter the virus titer (10.1152/ajplung.00352.2016, 10.1371/journal.ppat.1008506, 10.3389/fimmu.2020.00697). Discrepancies in the results may be explained by the dose-dependent effects of androgens, the type of the viruses or infected cells. Furthermore, the timing of the androgen treatments during infection can also influence the observed effects of the sex hormones as it was described previously in a bird model (http://www.jstor.org/stable/3067591).

In our paper we focused on the effects of androgens on the type I IFN responses or pDC functions. Thus, we did not mention the direct effect of testosterone on viral replication that is not completely clear based on the current literature data.

 „Line 673: Immunosuppressive agents can inhibit abnormal viral replication?. Please explain its roles.”

We thank for the Reviewer’s question. We described in the next paragraph that besides their immunomodulatory effects, these immunosuppressive drugs can also directly inhibit viral replication. In the revised manuscript we added more information about the antiviral effects of chloroquine to this paragraph, since the antiviral action of chloroquine is the most studied among the immunosuppressive agents. Please, find it in Line 713 in the revised version of the manuscript:

“For example, chloroquine interferes with different stages of the viral life cycle including viral entry, uncoating, assembly and budding. Via increasing endosomal pH chloroquine blocks virus-endosome fusion and is also able to inhibit posttranslational modifications of viral proteins by interfering with proteolytic processes (10.1016/j.tmaid.2020.101735, 10.1016/j.ijantimicag.2020.105938).”

We would like to say thank again the Reviewer for the valuable comments and recommendations, which helped us to improve the quality of our paper.

Reviewer 4 Report

This review is interesting and generally well written. Only few points deserve to be improved. In particular: 

Introduction: It deserves to behighlighted that, although at beginning of the pandemic, SARS-CoV2 was accounted as respiratory virus (which is true) that causes respiratory complications, in the last two years come out that this virus can also cause non respiratory complications including fertility and neuronal complications ( as recently reviewed PMID: 3497220635114008 and 35943095)

It would be helpful inserting a table at the end of each paragraph summarizing the studies analysed by the authors in that paragraph 

Author Response

Response to Reviewer 4.:

We are very grateful to the Reviewer for the time and effort taken to evaluate our manuscript and to help improve the quality of our paper. Based on the Reviewer’s comments and thoughtful suggestions we made changes to the manuscript. Please see our detailed response below.

 „Introduction: It deserves to behighlighted that, although at beginning of the pandemic, SARS-CoV2 was accounted as respiratory virus (which is true) that causes respiratory complications, in the last two years come out that this virus can also cause non respiratory complications including fertility and neuronal complications ( as recently reviewed PMID: 34972206, 35114008 and 35943095)”

We appreciate the Reviewer constructive suggestions, based on which we inserted a new paragraph in the Introduction section as follows:

“Besides the well-known respiratory pathology, various extrapulmonary manifestations of COVID-19 have already been reported highlighting the involvement of cardiovascular, genitourinary, gastrointestinal and central nervous system as well as the skin (10.1038/s41591-020-0968-3). The multi-organ involvement can be manifested by various symptoms including thrombotic complications, myocardial dysfunction, arrhythmia, acute coronary syndromes, acute kidney injury, gastrointestinal symptoms, hepatocellular injury, hyperglycemia and ketosis, neurologic illnesses, ocular symptoms, dermatologic complications, preeclampsia and fertility problems (10.1038/s41591-020-0968-3, 10.1093/brain/awab421, 10.1111/apm.13210, 10.1097/HJH.0000000000003213,, 10.1038/s42255-022-00554-4, 10.1016/j.jhep.2020.05.002).

In addition, those patients who recovered from COVID-19 may suffer from post-COVID-19 syndrome, which negatively affects their quality of life for months after recovery. The post-COVID-19 syndrome is characterized by a wide variety of clinical symptoms including pulmonary embolism, deep vein thrombosis, acute myocardial infarction, depression, anxiety, myalgia, dyspnea, fatigue, defects in memory and concentration and a variety of neuropsychiatric syndromes (10.1038/s41590-021-01104-y, 10.3390/vaccines9050497, 10.1007/s12016-021-08848-3, 10.2174/1871523021666220328115818).

In general the incidence of post-COVID-19 syndrome is about 10-35 %; however, this rate can reach up to 85% for those patients who required hospitalization during acute SARS-CoV-2 infection (10.3390/vaccines9050497).”

„It would be helpful inserting a table at the end of each paragraph summarizing the studies analysed by the authors in that paragraph „

We thank the Reviewer’s comment but we think that inserting a table at the end of each paragraph would unnecessarily break up the text and interrupt the flow of the content. In addition, the paragraphs not only contain recent articles closely related to the given topic, but also include earlier references for explanatory reasons. Formerly, we also planned to generate tables summarizing the recent findings on the features and functions of pDCs in COVID-19; however, previous reviews have already made this form of the communication of literature data. Therefore, we do not wish to follow this suggestion of the Reviewer.

However, based on the Reviewer’s comment we decided to indicate those references, which contain summarizing tables with literature data about the most extensively studied topics regarding COVID-19 such as the feature of IFN responses and pDCs in COVID-19, at the end of the appropriate paragraphs as follows:

Line 174-175: “A recent review with an excellent summarizing table provided a thorough overview of literature data on the features of IFN response in COVID-19 patients (10.3349/ymj.2021.62.5.381).”

Line 287-290: “The features of pDCs in COVID-19 are extensively reviewed in a recent paper [https://doi.org/10.1016/j.celrep.2022.111148], which thoroughly details the positive correlation between pDC function and COVID-19 severity, and provides a summarizing table about the observations regarding the fate of pDCs during COVID-19.”

We would like to express again our appretiation for the the Reviewer’s valuable comments, which helped us to improve the quality of our paper.